# C-Reactive Protein Levels Predict Improvement in the Liver Functional Reserve by Long-Term Rifaximin Treatment

**DOI:** 10.3390/diseases13100331

**Published:** 2025-10-08

**Authors:** Kensuke Kitsugi, Kazuhito Kawata, Go Murohisa, Yashiro Yoshizawa, Masaharu Kimata, Yosuke Kobayashi, Shuhei Unno, Hidenao Noritake, Takeshi Chida, Yoshisuke Hosoda

**Affiliations:** 1Department of Gastroenterology, Seirei Hamamatsu General Hospital, Shizuoka 430-8558, Japan; murohisa@sis.seirei.or.jp (G.M.); yashiro1224@sis.seirei.or.jp (Y.Y.); masa-kimata@sis.seirei.or.jp (M.K.); yosuke.k@sis.seirei.or.jp (Y.K.); s.unno@sis.seirei.or.jp (S.U.); yhosoda@sis.seirei.or.jp (Y.H.); 2Department of Internal Medicine II, Hamamatsu University School of Medicine, Shizuoka 431-3192, Japan; kawata@hama-med.ac.jp (K.K.); noritake@hama-med.ac.jp (H.N.); tchida@hama-med.ac.jp (T.C.)

**Keywords:** hepatic encephalopathy, inflammation, serum albumin

## Abstract

Objectives: Rifaximin is a non-absorbable antibiotic that has an efficacy for hepatic encephalopathy (HE). We previously demonstrated that rifaximin improved liver functional reserve, but this was a single-center study with a limited number of cases, and there were few cases of long-term use. Here, we conducted a multicenter study to evaluate the efficacy of long-term rifaximin administration on the liver functional reserve. Methods: A multicenter retrospective study was conducted on cirrhotic patients who received rifaximin for more than 12 months. We evaluated the efficacy of long-term rifaximin administration on the liver functional reserve. Results: A total of 65 cirrhotic patients were enrolled. Administration of rifaximin for 12 months significantly improved the Child–Pugh score (CPS) and albumin–bilirubin (ALBI) score. Regarding the parameters of the CPS, albumin scores significantly improved in addition to HE scores at 12 months. Univariate and multivariate analysis revealed that high C-reactive protein (CRP) levels (>0.69 mg/dL) at baseline were the predictive factor for improvement in the liver functional reserve. Conclusions: This study suggests that long-term rifaximin administration may improve the liver functional reserve in cirrhotic patients through improvement in albumin levels. CRP levels predict improvement in the liver functional reserve.

## 1. Introduction

Hepatic encephalopathy (HE) is a common complication in cirrhotic patients and the occurrence of HE deteriorates their prognosis, with a mortality rate within 1 year of 64% and within 5 years of 85% [1,2]. Therefore, it is important to improve HE and prevent its recurrence aspects by managing cirrhosis. The exacerbation of the liver functional reserve is correlated with poor prognosis. The liver functional reserve also affects the selection of treatment for hepatocellular carcinoma (HCC), and patients with poor a liver functional reserve are often forced to abandon HCC treatment. Therefore, the development of therapeutics focused on improving the liver functional reserve is an urgent issue.

Rifaximin is an oral antibiotic with minimal gastrointestinal absorption and broad-spectrum covering Gram-positive, Gram-negative, aerobic, and anaerobic bacterial flora via the inhibition of RNA synthesis [3]. Rifaximin inhibits ammonia-producing enteric bacteria and is recommended in practice guidelines in Japan, Europe, and the United States [4,5]. Moreover, rifaximin was demonstrated to prevent other complications of cirrhosis such as esophagogastric variceal bleeding, spontaneous bacterial peritonitis, and hepatorenal syndrome [6,7], or prolong overall survival [8].

Systemic inflammation plays a crucial role in cirrhosis progression. Patients with advanced cirrhosis have an elevated systemic inflammatory status [9]. Endotoxins and inflammatory cytokines are involved in the progression of cirrhosis [10], and endotoxemia is associated with the incidence and severity of HE [11]. In addition, several inflammation-based markers, such as the C-reactive protein (CRP) and neutrophil-to-lymphocyte ratio, can predict outcomes in cirrhotic patients [12,13]. Therefore, targeting the patient’s systemic inflammatory status seems to be beneficial. Rifaximin was also demonstrated to reduce serum levels of endotoxins and inflammatory cytokines [14]. Therefore, rifaximin is expected to improve the liver functional reserve through the amelioration of serum endotoxins and inflammatory cytokines. However, there are few reports investigating the efficacy of rifaximin on the liver functional reserve.

We previously reported that rifaximin improved the Child–Pugh score (CPS) at both 3 and 12 months [15]. In our previous study, rifaximin improved CRP levels and the CRP-to-albumin ratio, suggesting that the improvement in systemic inflammation by rifaximin administration may improve the liver functional reserve. However, this was a single-center study and there were few cases of long-term administration, making it difficult to evaluate the predictive factors for improvement in the liver functional reserve. Here, we conducted a multicenter retrospective study to evaluate the efficacy of long-term rifaximin administration on the liver functional reserve and the predictive factors for improvement in the liver functional reserve.

## 2. Materials and Methods

### 2.1. Patients

This was a retrospective observation study. The study population consisted of cirrhotic patients with overt HE who received 1200 mg of rifaximin per day (400 mg three times per day) for 12 months at Seirei Hamamatsu General Hospital and Hamamatsu University Hospital between April 2017 and November 2024. The exclusion criteria are as follows: (1) aged < 18 years, (2) discontinuation of rifaximin within 12 months due to death, side effects, or other reasons, (3) administration of other drugs for HE such as branched-chain amino acids (BCAAs), synthetic disaccharide, zinc preparation, and levocarnitine during the observation period, (4) performance of balloon-occluded retrograde transvenous obliteration (BRTO) during the observation period, and (5) patients with physical examination and imaging findings suggestive of infection at the rifaximin administration. Most of the excluded cases were deaths due to liver failure or concomitant HCC (60 cases), followed by cases with the administration of other drugs for HE during the observation period (21 cases), cases performing BRTO during the observation period (3 cases), and cases with discontinuation due to side effects (2 cases). The remaining 65 patients were enrolled in this study (Figure 1). This study was approved by the Ethics Committee of the Hamamatsu University Hospital (Ethics Approval Number: 24-275). Due to the retrospective nature of the study, informed consent was obtained through an opt-out option on the website.

### 2.2. Evaluations

The clinical and demographic characteristics of patients and laboratory data at baseline and 12 months after the administration of rifaximin were collected from medical records. The classification of HE was based on the West Haven criteria [16]. Recurrence of HE was defined as grade 2 or higher. The presence of ascites was evaluated by ultrasound or computed tomography examination. The liver functional reserve was evaluated using the CPS and albumin–bilirubin (ALBI) score. The CPS was calculated based on the severity of hepatic encephalopathy, ascites, total bilirubin, albumin, and prothrombin time. Cases without ascites while receiving diuretics were assessed as no ascites in the evaluation of the CPS. Patients were classified into three groups based on their CPS as follows: Child–Pugh class A (5–6), class B (7–9), and class C (10–15) [17]. The ALBI score was calculated as follows: (log10 {Total-bilirubin [μmol/L] × 17.1} × 0.66) + (Albumin [g/L] × −0.085). ALBI grades 1, 2, and 3 correspond to ALBI scores of ≤−2.60, <−2.60 to ≤−1.39, and >−1.39, respectively [18]. The change ratio was defined as the ratio of the post-treatment levels to the pre-treatment levels.

### 2.3. Statistical Analyses

Data on patient characteristics are presented as numbers for categorical data and medians and interquartile range as continuous variables. Differences between paired groups were analyzed using the Wilcoxon signed-rank test. The Mann–Whitney U test was used to compare independent samples. Categorical variables were compared between independent samples using Fisher’s exact test. Propensity scores were estimated by a logistic regression model with age. Matching between patients in the liver functional reserve improvement group and non-improvement group was performed according to the nearest neighbor method using a 0.2-width caliper and a 1:1 ratio. Spearman’s rank correlation coefficients (rs) were calculated to compare the variables. The accuracy for the outcome prediction was evaluated by the area under the receiver operator characteristic curve (AUROC), and the best cut-off value was calculated using the Youden Index. Univariate and multivariate analyses were performed using a logistic regression model for predicting the improvement in the liver functional reserve. All analyses were performed using EZR version 1.68; a modified version of the R commander was designed to add statistical functions frequently used in biostatistics [19]. A *p*-value of < 0.05 was considered statistically significant.

## 3. Results

### 3.1. Patient Characteristics

The baseline patient characteristics are summarized in Table 1. The median age of patients at rifaximin administration was 69 years. Males comprised 38 cases (59%), while females comprised 27 cases (41%). The most common etiology of cirrhosis was alcohol-associated liver disease (ALD) with 27 cases (42%), followed by metabolic dysfunction-associated steatohepatitis (MASH) with 14 (21%), hepatitis C virus (HCV) with 11 (17%), autoimmune with 8 (12%), and other causes. At baseline, 30 cases (46%) had ascites, 24 cases (37%) had esophagogastric varices, and 17 cases (26%) had HCC. Although five cases had a history of esophageal varix rupture, and three had a history of spontaneous bacterial peritonitis (SBP), none of the patients had esophageal varices rupture or SBP during the observation period. No cases received antibiotics other than rifaximin during the observation period. Among the cases with HCC, 13 cases (76%) underwent HCC treatment during the observation period. Five cases received molecular targeted agents, four cases underwent transcatheter arterial chemoembolization (TACE), two cases underwent radiofrequency ablation (RFA), and two cases received combined therapy with TACE and RFA. Almost all cases (97%) received concomitant medications at rifaximin administration, and more than half of the cases received BCAAs or synthesized disaccharides. Regarding the liver functional reserve, 95% of cases were classified as Child–Pugh class B or C, and 98% were ALBI grade 2 or 3. The median levels of ammonia was 106 μg/dL. Serum bilirubin level was normal, and prothrombin time and serum albumin levels were low. Inflammatory markers included normal white blood cell count and slightly elevated CRP levels.

### 3.2. Efficacy on Serum Ammonia Levels and Hepatic Encephalopathy

As shown in Figure 2a, serum ammonia levels significantly decreased after rifaximin administration, from 106 μg/dL at baseline to 59 μg/dL at 12 months (*p* < 0.001). The recurrence of HE was observed in 12 cases during the observation period, and the cumulative recurrence rate was 18.5% (Figure 2b).

### 3.3. Efficacy on Liver Functional Reserve

We next investigated the efficacy of long-term rifaximin administration on the liver functional reserve. The CPS improved in 49 cases (76%), remained unchanged in 10 cases (15%), and deteriorated in 6 cases (9%) at 12 months. As shown in Figure 3a, the median CPS value significantly improved from nine at baseline to seven at 12 months (*p* < 0.001). The Child–Pugh class improved in 32 cases (49%) (Figure 3b). The improvement in the Child–Pugh class was observed in 37% of class B. In cases with Child–Pugh class C, the median CPS value significantly improved (*p* < 0.001), and the improvement in the Child–Pugh class was observed in 75% of cases (Figure 4a,b). Regarding the ALBI score, improvement was observed in 39 cases (60%), whereas it remained unchanged in 1 case (2%) and deteriorated in 25 cases (38%). As shown in Figure 3c, the median ALBI score significantly improved from −1.47 at baseline to −1.77 at 12 months (*p* = 0.020). The ALBI grade improved in 12 cases of grade 3 (18%), while there were no cases of improvement in grade 2 (Figure 3d). The median ALBI score significantly improved (*p* = 0.001), and the improvement in the ALBI grade was observed in 46% of cases with ALBI grade 3 (Figure 4c,d).

We conducted further investigations to examine which parameters contributed to the improvement in the CPS and ALBI score. We evaluated the changes in the parameters of the CPS and ALBI score (HE, ascites, total bilirubin, albumin, and prothrombin time) after rifaximin administration. Figure 5a demonstrates the distribution of the five parameters that consist of the CPS at baseline and 12 months, whereas Figure 5b demonstrates the changes in albumin levels that consist of the ALBI score. As expected, HE scores demonstrated a significant improvement with rifaximin administration (Figure 5a). Other than the HE scores, there was a significant improvement in the albumin scores and levels (Figure 5a,b), whereas no significant changes were observed in the bilirubin scores and levels (1.6 mg/dL at baseline to 1.5 mg/dL at 12 months, *p* = 0.842). Moreover, the change ratio of albumin levels following rifaximin administration demonstrated a significant correlation with the change ratio of the CPS (rs = −0.579, *p* < 0.001) and ALBI score (rs = 0.947, *p* < 0.001).

### 3.4. Investigation of the Predictive Factors for Improvement in the Liver Functional Reserve

The baseline patient characteristics of the liver functional reserve improvement group and non-improvement group are summarized in Table 2. A significant difference was observed in the proportion of concomitant ascites and HCC (ascites, *p* = 0.012; HCC, *p* = 0.022). Moreover, serum albumin and ammonia levels were significantly lower, and CRP levels were significantly higher in the improvement group (albumin, *p* = 0.001; ammonia, *p* = 0.036; CRP, *p* = 0.005). After the patients in the improvement group were matched to the non-improvement group by age, a significant difference was observed in the proportion of HCV-related cirrhosis and concomitant ascites (HCV, *p* = 0.045; ascites, *p* = 0.029). As before matching, serum albumin and ammonia levels were significantly lower, and CRP levels were significantly higher in the improvement group (albumin, *p* = 0.004; ammonia, *p* = 0.029; CRP, *p* = 0.011). ROC analysis showed that serum albumin and CRP levels demonstrated moderate abilities to predict improvement in the liver functional reserve, whereas serum ammonia levels demonstrated a low ability (Figure 6). The cut-off values for serum albumin, ammonia, and CRP were 3.3 g/dL, 125 μg/dL, and 0.69 mg/dL, respectively.

We next performed univariate and multivariate analyses to identify the predictive factors for improvement in the liver functional reserve in the entire cohort (Table 3). Univariate analysis revealed that age < 65 years (*p* = 0.018), presence of ascites (*p* = 0.009), absence of HCC (*p* = 0.016), albumin level < 3.3 g/dL (*p* = 0.030), ammonia level < 125 μg/dL (*p* = 0.027), and CRP level > 0.69 mg/dL (*p* = 0.011) were significantly associated with improvement in the liver functional reserve. Multivariate analysis revealed that a CRP level of >0.69 mg/dL (*p* = 0.018) was independently associated with improvement in the liver functional reserve. Furthermore, same as albumin, the change ratio of CRP levels following rifaximin administration demonstrated a significant correlation with the change ratio of CPS (rs = 0.508, *p* < 0.001) and ALBI score (rs = −0.635, *p* < 0.001).

## 4. Discussion

In the present study, we evaluated the efficacy of long-term rifaximin administration on the liver functional reserve and demonstrated that rifaximin administration for 12 months improved the CPS and ALBI score, mainly due to the improvement in serum albumin levels. Moreover, the investigation of the prognostic factor suggested that patients with high CRP levels may benefit from improvement in the liver functional reserve. Cirrhosis is regarded as the terminal state of chronic liver disease and has a poor prognosis with the one-year survival rate of patients with Child–Pugh class C approximately 45% [20]. The only curative treatment for decompensated cirrhosis is liver transplantation, but this has some problems, such as the shortage of a donor, and the development of therapeutics focused on improving the liver functional reserve is an urgent issue. In this study, 75% of the patients with Child–Pugh class C improved to class B, and 46% of the patients with ALBI grade 3 improved to grade 1 or 2. Therefore, we believe this study is significant in that it demonstrates the possibility that rifaximin may contribute to improvement in the liver functional reserve even in patients with poor a liver functional reserve.

Our previous report demonstrated that rifaximin improved the liver functional reserve in cirrhotic patients by ameliorating CRP and serum albumin [15]. However, the number of cases of long-term rifaximin administration was limited, making it difficult to conduct a detailed examination. Likewise with our previous report, the current study suggested that albumin and CRP were deeply involved with improvement in the liver functional reserve by rifaximin administration. Albumin and CRP are serological inflammation-based markers and involved in the progression of cirrhosis [21]. Among the etiologies of cirrhosis, inflammation plays an important role in the progression of ALD [22]. Kalambokis et al. reported that 8 weeks of rifaximin administration improved the CPS and model for end-stage liver disease scores in patients with ALD [23]. In our study, ALD was the most common etiology of cirrhosis. However, ALD was not a predictor of improvement in the liver functional reserve, and rifaximin also improved the liver functional reserve in patients with other etiologies, such as HCV and MASH. Our study suggests that rifaximin may be effective in the improvement in the liver functional reserve in cases of cirrhosis other than ALD.

Moreover, the increase in the number of cases in the multicenter study enabled the investigation of predictive factors for improvement in the liver functional reserve, and we identified serum CRP levels as a predictive factor. Systemic inflammatory response plays a crucial role in the progression of cirrhosis, and CRP is a surrogate marker for systemic inflammatory response [10]. High CRP levels in cirrhotic patients are associated with poor prognosis [12,24]. Not only baseline CRP levels, but also the reduction in CRP levels following treatments such as antibiotics has been reported as a predictor of prognosis in cirrhotic patients [25,26]. Correspondingly, our study demonstrated a significant correlation between the improvement ratio of CRP and the improvement ratio of the liver functional reserve. These results suggest that CRP may be a key target for improvement in the liver functional reserve in cirrhotic patients. Inflammatory cytokines such as interleukin-6 (IL-6) and tumor necrosis factor-alpha (TNF-α) also play important roles in the progression of cirrhosis [9,10] and are involved in the complications such as HE and HCC [27,28]. Several reports demonstrated that rifaximin also improved these inflammatory cytokines [23,29]. Moreover, these inflammatory cytokines promote the production of CRP [30,31]. Therefore, inflammatory cytokines and CRP are closely related. Measuring inflammatory cytokines in daily clinical practice is difficult, whereas CRP is inexpensive and easily available.

In this study, the improvement in the CPS and ALBI score by rifaximin was strongly associated with improvement in serum albumin levels. Moreover, rifaximin also improved the liver functional reserve in cases with ascites. Because albumin is mainly synthesized in the liver, cirrhotic patients exhibit hypoalbuminemia and reduction in colloid osmotic pressure, consequently developing ascites [32]. Therefore, albumin and ascites are important therapeutic targets in cirrhotic patients. Kawaratani et al. reported that long-term rifaximin treatment improved serum albumin levels [33]. Moreover, dysbiosis is associated with refractory ascites in cirrhotic patients, and rifaximin improves gut microbial changes and intestinal permeability, thereby inhibiting cirrhotic complications including ascites [14,21,34]. Furthermore, albumin is also correlated with systemic inflammation. There is a negative correlation between serum albumin levels and blood endotoxin levels [35], and the CRP-to-albumin ratio correlates with poor prognoses in cirrhotic patients [36]. Our previous study also demonstrated that rifaximin improved the CRP-to-albumin ratio [15]. As mentioned above, rifaximin demonstrated an inhibitory effect on systemic inflammation in previous reports [23,29]. These results suggest that rifaximin may improve the liver functional reserve by inhibiting systemic inflammation, thereby improving serum albumin levels and ascites. However, further investigations such as the evaluation of inflammatory cytokines, endotoxins, and gut microbiota are required.

An important point of this study is that rifaximin improved the liver functional reserve even in cases of poor liver functional reserves such as Child–Pugh class C and ALBI grade 3. Also in the previous report, rifaximin is effective and safe even in cases of Child–Pugh class C and improved serum albumin levels [33]. Therefore, it is suggested that rifaximin may be effective in cases where systemic inflammation is strongly involved. Cirrhotic patients with high CRP levels may benefit from improvement in the liver functional reserve with rifaximin, regardless of Child–Pugh class and ALBI grade. We believe that rifaximin administration should be actively considered in cirrhotic patients with the presence of systemic inflammation, specifically in patients with high CRP levels, low albumin levels, or ascites.

This study has some limitations. First, although this was a multicenter retrospective study, the sample size was still small. A prospective randomized controlled trial with larger samples is required in the future. Second, this study included patients who were able to take rifaximin for more than 12 months to investigate its long-term efficacy on the liver functional reserve. In fact, most of the excluded cases were deaths due to liver failure or concomitant HCC. Therefore, the possibility cannot be excluded that selection bias may have resulted in the inclusion of many cases with favorable prognoses. Third, other drugs for HE treatment may affect the liver functional reserve. In this study, we excluded cases in which other drugs for HE treatment were newly administered within the observation period to examine the efficacy of rifaximin itself. However, we did not exclude cases in which other drugs for HE treatment had already been administered before rifaximin administration, and almost all cases were taking one or more drugs for HE treatment at the initiation of rifaximin administration. The long-term efficacy of these drugs in improving the liver functional reserve cannot be denied. Moreover, diuretics were added or reduced during the observation period in many cases. Therefore, the possibility that diuretics affected the liver functional reserve could not be excluded. Fourth, this study demonstrated the possibility that rifaximin improved the liver functional reserve, but the impact on prognosis could not be evaluated. Moreover, this study was limited to cirrhotic patients complicated with HE, and it would be desirable to investigate the efficacy of rifaximin on the liver functional reserve in all cirrhotic patients. Fifth, CRP levels can also change due to systemic infection. We excluded patients with symptoms or findings suggestive of infection at rifaximin administration. However, it was difficult to rule out systemic infections that were lacking typical symptoms.

## 5. Conclusions

This multicenter retrospective study demonstrated that long-term rifaximin administration was not only effective and safe but also contributed to improving the liver functional reserve, and serum CRP levels may be a predictor of the improvement in the liver functional reserve. Rifaximin may improve the liver functional reserve by inhibiting systemic inflammation, thereby improving serum albumin levels. Further investigation of molecular mechanisms through basic experiments are required. Nonetheless, we believe it is significant that this study demonstrated the potential of rifaximin to improve the liver functional reserve with appropriate patient selection.

## Figures and Tables

**Figure 1 diseases-13-00331-f001:**
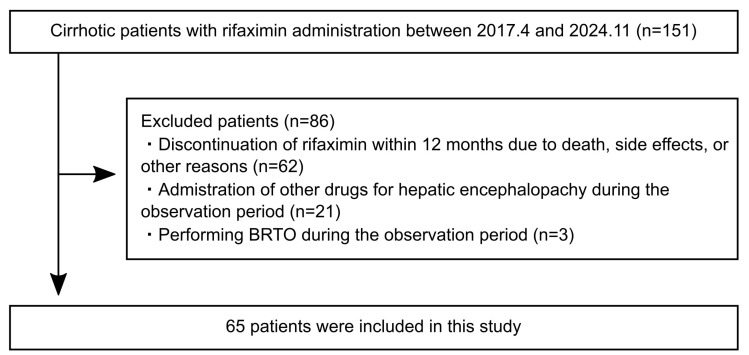
Flow diagram of the study. BRTO, balloon-occluded retrograde transvenous obliteration.

**Figure 2 diseases-13-00331-f002:**
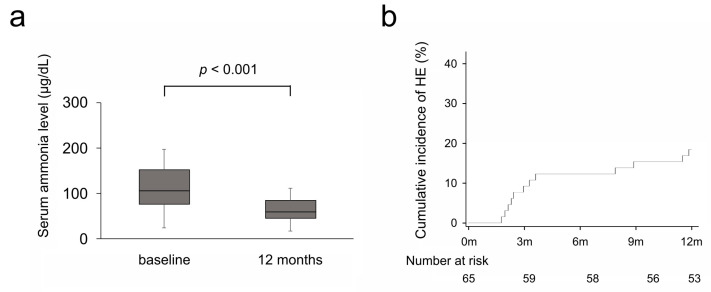
(**a**) Serum ammonia levels at baseline and 12 months. (**b**) Cumulative incidence of recurrence of HE. 0 m, 3 m, 6 m, 9 m, and 12 m indicate baseline, 3 months, 6 months, 9 months, and 12 months after rifaximin administration, respectively. HE, hepatic encephalopathy.

**Figure 3 diseases-13-00331-f003:**
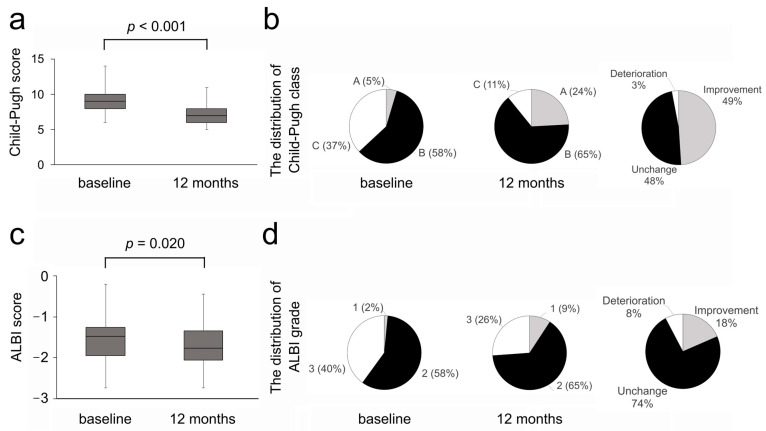
(**a**) Child–Pugh score at baseline and 12 months. (**b**) The distribution of Child–Pugh class at baseline and 12 months and changes in Child–Pugh class at 12 months. (**c**) ALBI score at baseline and 12 months. (**d**) The distribution of ALBI grade at baseline and 12 months and changes in ALBI grade at 12 months. ALBI, albumin–bilirubin.

**Figure 4 diseases-13-00331-f004:**
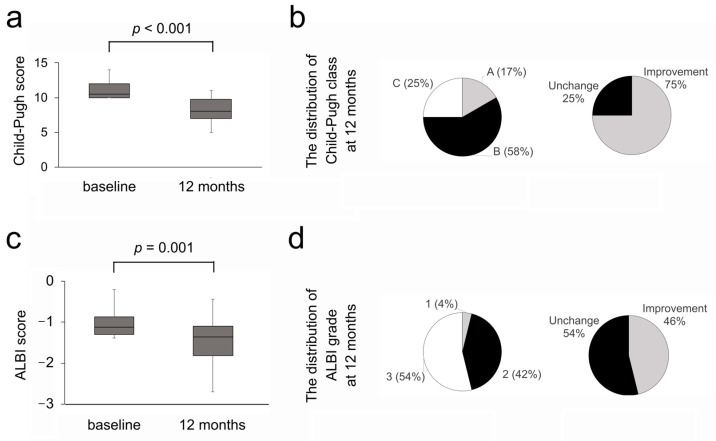
(**a**) Child–Pugh score at baseline and 12 months in cases with Child–Pugh class C. (**b**) The distribution of Child–Pugh class and changes in Child–Pugh class at 12 months in cases with Child–Pugh class C. (**c**) ALBI score at baseline and 12 months in cases with ALBI grade 3. (**d**) The distribution of ALBI grade and changes in ALBI grade at 12 months in cases with ALBI grade 3. ALBI, albumin–bilirubin.

**Figure 5 diseases-13-00331-f005:**
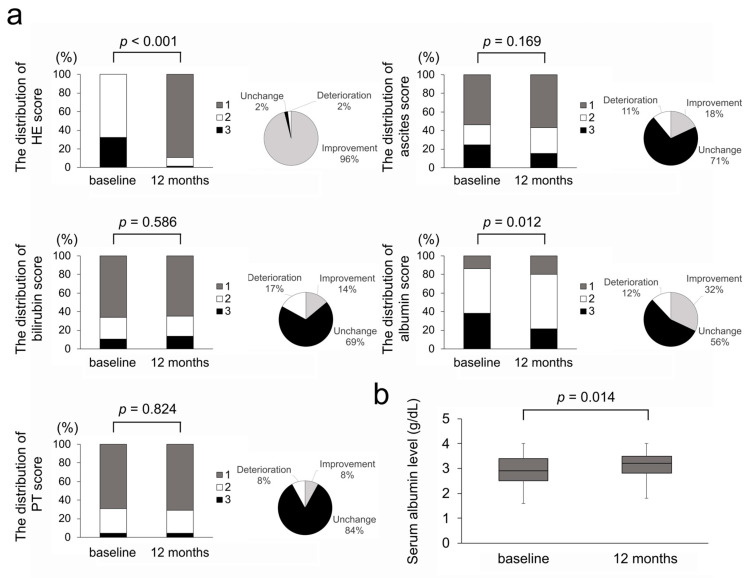
(**a**) The distribution of scores in parameters of the Child–Pugh score at baseline and 12 months and changes in each parameter at 12 months. (**b**) Serum albumin levels at baseline and 12 months. HE, hepatic encephalopathy; PT, prothrombin time.

**Figure 6 diseases-13-00331-f006:**
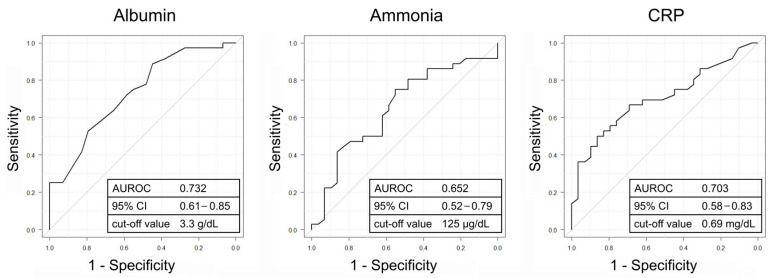
ROC analysis of the baseline values of serum albumin, ammonia, and CRP to predict improvement in the liver functional reserve. The diagonal line represents the behavior of a random classifier. AUROC, the area under the receiver operating characteristic; CI, confidence interval; CRP, C-reactive protein.

**Table 1 diseases-13-00331-t001:** Baseline patient characteristics.

Variable	Results
Age [years]	69 (60–76)
Sex	
Male gender, n (%)	38 (59)
Female gender, n (%)	27 (41)
Etiology, n (%)	
Alcohol	27 (42)
MASH	14 (21)
HCV	11 (17)
Autoimmune	8 (12)
Cryptogenic	3 (5)
HBV	2 (3)
Complication, n (%)	
Ascites	30 (46)
Esophagogastric varices	24 (37)
HCC	17 (26)
Concomitant drug *, n (%)	
BCAA	58 (89)
Lactulose or Lactitol	42 (65)
Levocarnitine	28 (43)
Zinc preparation	7 (11)
Without concomitant drug	2 (3)
Liver functional reserve	
Child–Pugh score	9 (8–10)
Child–Pugh class A:B:C, n (%)	3:38:24 (5:58:37)
ALBI score	−1.47 (−1.95, −1.26)
ALBI grade 1:2:3, n (%)	1:38:26 (2:58:40)
Laboratory data	
Total bilirubin [mg/dL]	1.6 (1.1–2.3)
Prothrombin time [%]	67 (52–78)
Albumin [g/dL]	2.9 (2.5–3.4)
Ammonia [μg/dL]	106 (76–152)
WBC [/μL]	4695 (3462–6035)
CRP [mg/dL]	0.36 (0.16–1.01)

Data are given in median and interquartile range for age, liver functional reserve, and laboratory data are added. * There is duplication. Abbreviations: ALBI, albumin–bilirubin; BCAA, branched-chain amino acid; CRP, C-reactive protein; HBV, hepatitis B virus; HCC, hepatocellular carcinoma; HCV, hepatitis C virus; MASH, metabolic dysfunction-associated steatohepatitis; WBC, white blood cell.

**Table 2 diseases-13-00331-t002:** Baseline patient characteristics in the liver functional reserve improvement group and non-improvement group.

	Before Matching			After Matching		
Variable	Improvement Group (n = 36)	Non-Improvement Group (n = 29)	*p*-Value	Improvement Group (n = 21)	Non-Improvement Group (n = 21)	*p*-Value
Age [years]	65 (60–75)	75 (67–78)	0.074	71 (64–76)	71 (65–76)	0.980
Sex			0.800			1.000
Male gender, n (%)	22 (61)	16 (55)		12 (57)	12 (57)	
Female gender, n (%)	14 (39)	13 (45)		9 (43)	9 (43)	
Etiology, n (%)						
Alcohol	17 (47)	10 (35)	0.324	9 (43)	9 (43)	1.000
MASH	6 (17)	8 (27)	0.367	4 (19)	6 (29)	0.719
HCV	8 (22)	3 (10)	0.320	7 (34)	1 (4)	0.045
Autoimmune	4 (11)	4 (14)	1.000	1 (4)	2 (10)	1.000
Cryptogenic	1 (3)	2 (7)	0.582	0 (0)	2 (10)	1.000
HBV	0 (0)	2 (7)	0.195	0 (0)	1 (4)	1.000
Complication, n (%)						
Ascites	22 (61)	8 (28)	0.012	14 (67)	6 (29)	0.029
Esophagogastric varices	15 (42)	9 (31)	0.444	9 (43)	6 (29)	0.520
HCC	5 (14)	12 (41)	0.022	4 (19)	8 (38)	0.306
Concomitant drug *, n (%)						
BCAA	31 (86)	27 (93)	0.447	18 (86)	19 (91)	1.000
Lactulose or Lactitol	20 (56)	22 (76)	0.119	15 (71)	16 (76)	1.000
Levocarnitine	13 (36)	15 (52)	0.221	12 (57)	10 (48)	0.758
Zinc preparation	5 (14)	2 (7)	0.447	3 (14)	1 (5)	0.606
Without concomitant drug	2 (6)	0 (0)	0.498	1 (5)	0 (0)	1.000
Laboratory data						
Total bilirubin [mg/dL]	1.6 (1.1–2.3)	1.7 (1.0–2.1)	0.947	1.5 (0.9–2.4)	1.7 (0.9–2.1)	0.696
Prothrombin time [%]	66 (50–77)	68 (58–79)	0.417	62 (50–77)	68 (57–79)	0.428
Albumin [g/dL]	2.7 (2.4–3.1)	3.2 (2.8–3.6)	0.001	2.6 (2.2–3.0)	3.2 (2.8–3.6)	0.004
Ammonia [μg/dL]	98 (74–127)	127 (92–161)	0.036	90 (69–122)	127 (92–198)	0.029
WBC [/μL]	4830 (3065–6205)	4680 (3640–5550)	0.941	4620 (3110–5230)	4380 (3620–5040)	0.920
CRP [mg/dL]	0.62 (0.22–1.73)	0.25 (0.10–0.41)	0.005	0.69 (0.25–2.00)	0.25 (0.10–0.44)	0.011

Data are given in median and interquartile range for age and laboratory data are added. * There is duplication. Abbreviations: BCAA, branched-chain amino acid; CRP, C-reactive protein; HBV, hepatitis B virus; HCC, hepatocellular carcinoma; HCV, hepatitis C virus; MASH, metabolic dysfunction-associated steatohepatitis; WBC, white blood cell.

**Table 3 diseases-13-00331-t003:** Univariate and multivariate analysis of predictive factors in baseline patient characteristics for improvement in the liver functional reserve in the entire cohort.

Variable	Categories	Univariate Analysis			Multivariate Analysis		
		OR	95% CI	*p*-Value	OR	95% CI	*p*-Value
Age [years]	<65 vs. ≥65	3.83	1.26–11.60	0.018	3.51	0.82–15.00	0.091
Sex	Male vs. female	1.28	0.47–3.44	0.629			
Etiology							
	Alcohol vs. others	1.70	0.62–4.65	0.302			
	MASH vs. others	0.53	0.16–1.74	0.291			
	HCV vs. others	2.48	0.59–10.40	0.214			
	Autoimmune vs. others	0.78	0.18–3.44	0.744			
Complication							
Ascites	Presence vs. absence	4.12	1.44–11.80	0.009	1.47	0.39–5.57	0.569
Esophagogastric varices	Presence vs. absence	1.59	0.57–4.44	0.379			
HCC	Presence vs. absence	0.23	0.07–0.76	0.016	0.18	0.03–1.12	0.066
Concomitant drug at baseline							
BCAA	With vs. without	0.46	0.08–2.56	0.375			
Lactulose or Lactitol	With vs. without	0.40	0.14–1.17	0.093			
Levocarnitine	With vs. without	0.53	0.20–1.43	0.208			
Zinc preparation	With vs. without	2.18	0.39–12.10	0.375			
Laboratory data							
Total bilirubin [mg/dL]	<1.6 vs. ≥1.6	1.64	0.61–4.42	0.332			
Prothrombin time [%]	>67 vs. ≤67	0.84	0.31–2.22	0.718			
Albumin [g/dL]	<3.3 vs. ≥3.3	3.27	1.12–9.54	0.030	3.34	0.86–13.00	0.082
Ammonia [μg/dL]	<125 vs. ≥125	3.20	1.14–8.99	0.027	2.32	0.62–8.75	0.213
WBC [/μL]	<4695 vs. ≥4695	0.88	0.33–2.36	0.802			
CRP [mg/dL]	>0.69 vs. ≤0.69	5.00	1.44–17.30	0.011	8.90	1.45–54.50	0.018

The cut-off value for total bilirubin, prothrombin time, and white blood cells were the median value of all cases, and for albumin, ammonia, and CRP were the value obtained by ROC analyses. Abbreviations: BCAA, branched-chain amino acid; CI, confidence interval; CRP, C-reactive protein; HCC, hepatocellular carcinoma; HCV, hepatitis C virus; MASH, metabolic dysfunction-associated steatohepatitis; OR, odds ratio; ROC, receiver operating characteristic; WBC, white blood cell.

## Data Availability

The data that support the findings of this study are available from the corresponding author upon reasonable request.

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
