# Peer review of "C-Reactive Protein Levels Predict Improvement in the Liver Functional Reserve by Long-Term Rifaximin Treatment"

_diseases, 2025, doi:10.3390/diseases13100331_

Round 1

Reviewer 1 Report

Comments and Suggestions for Authors

This is an interesting study exploring the possible efficacy of long-term rifaximin in managing liver disease, liver cirrhosis in particular, and improving liver reserve. The design of the study is appropriate. The research question sounds good, and the results are clearly presented. Limitations were a lot, but you defined them, and that is very important. I have no further comments to address. 

Reviewer 2 Report

Comments and Suggestions for Authors

This manuscript suggests that higher CRP contributes to improved liver function after rifaximin administration. While it seems to be an interesting result, it presents several significant issues.

  1. The sample size is too small to make conclusions. If conclusions are to be made based on these results, validation or propensity matching should be performed. Although age showed no statistically significant difference, younger patients appear to experience greater improvement in liver function. At least, background age should be adjusted for.
  2. What were the criteria for rifaximin administration? The criteria for administration and the duration of treatment should be clearly stated.
  3. Regarding hepatic encephalopathy (HE), were overt cases being studied? What criteria were used to define HE?
  4. How was the presence or absence of ascites assessed? Was diuretic use evaluated? How were cases without ascites despite diuretic therapy assessed?
  5. CRP levels can also change due to systemic infection. How is the presence or absence of infection assessed? Furthermore, if ascites is present, is bacterial peritonitis being ruled out? The results indicate that patients with high CRP and low albumin levels show easier improvement in liver function with RFX. Could this improvement in such cases be due to a preceding infection and the subsequent decline in liver function? Author mentioned no cases with SBP during the observation period. Were there any cases with a history of SBP prior to RFX administration, or cases treated with antibiotics during the observation period?

Reviewer 3 Report

Comments and Suggestions for Authors

The manuscript entitled "C-Reactive Protein Levels Predict the Improvement of Liver 2 Functional Reserve by long-Term Rifaximin Treatment" submitted by  Kensuke Kitsugi et al., is interesting and original. 

However there are different critical point to be addressed.

  • Introduction is really poor, the authors must extend the descritption of the topic
  • The results must be better described, it is not clear the specific results obtained between male and female patients. There are two table, but in the top is written only male gender, what does it mean ?? Have the authors analyzed female patients??
  • Concerning the results described, the authors must improve the dataset, also including negative results and in the graphs is not clear the standard error bar.
  • The authors must include a paragraph concerning the limit of this study, also because the number of patients is a small population.

Reviewer 4 Report

Comments and Suggestions for Authors

The article is devoted to the study of the effectiveness of long-term use of rifaximin on the functional reserve of the liver in cirrhotic patients. This work represents an important step in the field of hepatology, but it requires attention to several aspects in order to increase its scientific novelty and significance. Taking into account the comments below, the article has the potential to make a significant contribution to clinical practice in the treatment of cirrhosis and can be published in the journal Desease, however, it requires further development and in-depth study of the presented results.

1. The work clarifies the effect of rifaximin on liver reserves, but its novelty is limited, since it covers the already known properties of the drug, such as improving albumin and reducing encephalopathy. It would be necessary to consider in more detail the molecular mechanisms of action and try to identify new therapeutic approaches, for example, through integration with other drugs.

2. The multi-center approach adds static reliability, but the sample size (65 patients) is still small. Increasing the number of participants will add representativeness to the data obtained.

3. The use of a retrospective approach may limit the quality of the data obtained. It is necessary to add an analysis through prospective studies to strengthen the causal conclusions.

4. It is interesting to note the predictive value of the CRP level for improving the functional reserve of the liver. However, this observation requires a more detailed analysis and control of additional variables that may affect the results.

5. A discussion on the personalization of treatment for patients with high levels of CRP could complement the article by offering more specific recommendations for clinical practice.

6. It is important to improve the graphic part for a more visual presentation of the results, which will facilitate the perception of the material by a wide audience.

7. The conclusions should be expanded with additional information on the potential mechanisms of action of rifaximin and its long-term use, which would enhance the clinical relevance of the research.

Round 2

Reviewer 2 Report

Comments and Suggestions for Authors
  1. The sample number is considered critical and there must be information that convinces the reader. Among the 151 cases registered, 86 were excluded in this study. While 65 cases were included in the retrospective study, the fact that the number of exclusions exceeds the number of inclusion numbers suspects that lots of cases showing no improvement in liver function were excluded from the outset. If these are to be considered predictive factors for pre-treatment efficacy, accurate evaluation requires including cases that dropped out during the study. Should be clarify the exclusion cases more detailed. In addition, did all dropout cases have high CRP levels?
  2. Were the univariate and multivariate analyses performed based on matched cases? The description is unclear.

Author Response

Plese see the attachment.

Reviewer 3 Report

Comments and Suggestions for Authors

The authors improved the manuscript, it is ready to be accepted 

Reviewer 4 Report

Comments and Suggestions for Authors

Thank you for sending the edits.
